# Design, Synthesis and Herbicidal Activity of 1,2,4-Oxadiazole Compounds as Novel Light-Dependent Protochlorophyllide Oxidoreductase Inhibitors

**DOI:** 10.3390/molecules30193970

**Published:** 2025-10-03

**Authors:** Xiao Hu, Jing Miao, Yiyi Tian, Wennan Luo, Jixian Shang, Ruiyuan Liu, Huizhe Lu

**Affiliations:** 1Department of Chemistry, College of Science, China Agricultural University, Beijing 100193, China; s20233102246@cau.edu.cn (X.H.); mjing@cau.edu.cn (J.M.); tianyiyi2000@163.com (Y.T.); 18786103934@163.com (W.L.); zb20243100976@cau.edu.cn (J.S.); 2CAU-EastMab Joint Laboratory for Intelligent Screening and Creation of Active Molecules, China Agricultural University, Beijing 100193, China; 3CAS Key Laboratory of Pathogen Microbiology and Immunology, Institute of Microbiology, Chinese Academy of Sciences, Beijing 100101, China; liuruiyuan0924@126.com

**Keywords:** LPOR inhibitor, 1,2,4-oxadiazole compounds, herbicidal activity, molecular docking, crop safety, molecular dynamic simulation

## Abstract

Light-dependent protochlorophyllide oxidoreductase (LPOR, E.C.1.3.1.33) plays a crucial role in the biosynthesis of chlorophyll in plants. Therefore, inactivating LPOR can hinder the production of chlorophyll to achieve the effect of weed control. In this research, utilizing an active substructure splicing method, 20 new 1,2,4-oxadiazole compounds targeting LPOR were synthesized. Among them, compounds **5j**, **5k** and **5q** exhibited superior inhibitory efficacy in greenhouse herbicidal trials. In vitro enzyme activity assays indicated that **5q** significantly inhibited *Arabidopsis thaliana* LPOR (*At*LPOR), with an IC_50_ value of 17.63 μM. Furthermore, compound **5q** exhibited superior crop safety and holds potential application prospects for weed management in cotton. Molecular docking and dynamic simulations were employed to elucidate the binding mode and molecular mechanism of **5q** with *At*LPOR. These experimental and theoretical results indicate that **5q** is a promising candidate for the development of novel herbicides targeting LPOR.

## 1. Introduction

Weeds compete with economically important crops for plant growth resources, resulting in significant losses in agricultural production [1,2,3]. Consequently, several weed management strategies, including cultural [4], mechanical [5], biological [6,7] and chemical [8] approaches, have been applied in agricultural production. Among these, chemical control is the most effective for improving crop quantity and quality while reducing labor costs. In addition, the research and development of targeted pesticides is an inevitable trend and necessary to obtain high-efficiency and low-toxicity pesticides [9]. However, more than 300 herbicides have been applied in the world, whereas there are less than 20 kinds of targets [10]. So, the exploration of herbicides with new structures, which are aimed at novel target enzymes especially, will be beneficial to overcome resistance and some other problems so as to discover herbicides with a new mechanism of action (MOA) [11].

Light-dependent protochlorophyllide oxidoreductase (LPOR, E.C.1.3.1.33) emerged as a novel target protein for herbicide development in 2023 [12,13]. LPOR catalyzes the penultimate step using cofactor NADPH in the chlorophyll biosynthesis pathway: the photoreduction of protochlorophyllide (Pchlide) to chlorophyllide (Chlide) [14]. LPOR, as one of the very few light-driving enzymes, plays an indispensable role in photosynthesis [15]. The deficiency of LPOR will trigger a cascade of chloroplast developmental defects and photosynthetic pigment decline, leading to a seedling-lethal phenotype [16].

Heterocycles are important substructures that play a key role in pharmaceutical and agrochemical functions [17,18,19,20]. Notably, approximately three-quarters of herbicides and pesticides contain at least one heterocyclic ring [21]. The biological activity exhibited by heterocycles is attributed to their potential to bind to active sites or enzyme pocket structures with various enzymes through a wide range of intramolecular interactions such as van der Waals forces and hydrophobic bonds, hydrogen bonds and metal coordination bonds, making them important scaffolds in medicinal chemistry [22]. In addition, oxadiazole has great potential for use in discovering new pesticides, exemplified by the commercial pesticides metoprazolidone and oxadiazole, which have been used as corresponding pesticides and herbicides in agriculture [23,24].

In our previous work, through virtual screening and bioactivity testing, we obtained three small molecule inhibitors that targeted LPOR with relatively good activity [12,13]. In this study, in order to obtain novel inhibitors targeting LPOR, the previously mentioned active small fragments (heterocyclic oxadiazole, benzene ring, ether bonds) were ligated together to obtain the target compound, as shown in Figure 1. Based on the above analysis and the binding mode of the inhibitors, we designed and synthesized a series of new 1,2,4-oxadiazole compounds. Systematic postemergence herbicidal activity tests of these compounds were conducted. In addition, the *Arabidopsis thaliana* LPOR (*At*LPOR) inhibitory activity and crop safety of some representative compounds were evaluated. Furthermore, molecular docking studies and molecular dynamic simulation were performed to gain a deeper understanding of the inhibitory mechanism. This study suggests that 1,2,4-oxadiazole compounds may serve as potential lead compounds for the development of novel LPOR inhibitors.

## 2. Results and Discussion

### 2.1. Chemistry

Based on inhibitor binding mode analysis, the benzene ring can form π-cation interactions with residues, and the oxygen atom of symmetric bonds can act as a hydrogen bond receptor to form hydrogen bonds. The target compounds were designed based on the active substructure splicing method and synthesized according to Figure 1. It is worth mentioning that hydroxylammonium hydrochloride and Na_2_CO_3_ need to be dissolved in water first; then nitrile compounds were mixed with ethanol, and this solution was added to the former solution. We found that if the above two substances and nitrile compounds were only mixed with ethanol, sodium carbonate could not be dissolved, resulting in a lower yield at this step.

More importantly, we need to make sure that when synthesizing compound **2**, compound **1** is fully reacted; otherwise it will be difficult to separate compound 1 in the final purification step.

All target structures were confirmed by ^1^H NMR, ^13^C NMR and HRMS data and are listed in the Appendix A.

### 2.2. Herbicidal Activity Assay and SAR

Under greenhouse conditions, we evaluated the postemergence herbicidal activity of 20 target compounds against broadleaf weeds (Arabidopsis thaliana, AT), with SUT used as a positive control. As shown in Table 1 and Figure 2, the herbicidal activity of the target compounds varied significantly. As the dose is reduced, the survival rate of *Arabidopsis thaliana* increased. At a dose of 150 g ai/ha, most compounds caused scorching, necrosis and then the death of the tested weeds. But at a dose of 37.5 g ai/ha, most compounds showed less than 50% inhibition for the tested weeds. At the same time, we found that even at a dose of 37.3 g ai/ha, compounds **5j**, **5k** and **5q** showed good herbicidal activity against *Arabidopsis thaliana*, and inhibition was greater than 90% at a dose of 150 g ai/ha. Subsequently, we assessed the postemergence herbicidal activity of compounds **5j**, **5k** and **5q** against grass weed (Digitaria sanguinalis, DS). As shown in Appendix A, all compounds showed good herbicidal activity at a dose of 150 g ai/ha. In comparison, compound **5q** had the best herbicidal activity, even exceeding that of the control group SUT.

To explore the relationship between the chemical structure of the target compounds and their herbicidal efficacy, we analyzed the Structure–Activity Relationship (SAR). First, we found that the substituents on benzene had a better inhibition effect when they had electron-withdrawing groups. For example, at a dose of 75 g ai/ha, most compounds with electron-withdrawing groups attached to the phenyl ring had an inhibition rate higher than 50% for weeds, and the best inhibition effect was achieved when the substituents were NO_2_ (**5j**, **5k**). In contrast, even at a concentration of 150 g ai/ha, most compounds with electron-donor groups on the phenyl ring inhibit weeds by less than 50%. However, it is worth noting that when the substituent is OCH_2_Ph (**5q**), herbicidal activity is very good. Furthermore, the position of the substituents on the phenyl ring had minimal impact on herbicidal activity [5b and 5c (R = Cl), 5d and 5e (R = F), 5f and 5g (R = I), 5h and 5i (R = Br), **5j** and **5k** (R = NO_2_)].

**Table 1 molecules-30-03970-t001:** *Arabidopsis thaliana* herbicidal activity of compounds.

Compound		Dosage (g ai/ha)		
		37.5	75	150
**5a**	4-OCH_2_CH_3_	37.5 ± 1.8	50.3 ± 1.1	66.1 ± 4.5
**5b**	3-Cl	48.2 ± 1.6	62.3 ± 3.8	83.5 ± 1.5
**5c**	4-Cl	45.4 ± 3.1	67.5 ± 3.8	75.5 ± 0.3
**5d**	3-F	38.3 ± 2.1	68.3 ± 2.1	74.6 ± 1.3
**5e**	4-F	37.5 ± 3.0	56.4 ± 3.3	68.5 ± 2.1
**5f**	3-I	32.3 ± 1.9	53.0 ± 2.3	62.2± 1.7
**5g**	4-I	28.1 ± 3.1	57.4 ± 1.1	70.3 ± 1.8
**5h**	3-Br	45.3 ± 0.9	40.0 ± 0.8	53.8 ± 2.8
**5i**	4-Br	47.2 ± 1.0	57.7 ± 1.8	57.4 ± 1.1
**5j**	3-NO_2_	80.2 ± 1.4	92.6 ± 0.5	92.3 ± 2.6
**5k**	4-NO_2_	82.3 ± 2.6	98.6 ± 1.4	94.6 ± 1.9
**5l**	-	22.9 ± 0.8	27.4 ± 3.5	33.9 ± 2.8
**5m**	4-SCH_3_	23.0 ± 1.3	28.7 ± 2.9	55.2 ± 1.3
**5n**	4-CH_3_	16.1 ± 2.4	26.7 ± 0.8	40.4 ± 3.1
**5o**	4-OCH_3_	27.9 ± 1.5	35.6 ± 3.1	43.6 ± 1.4
**5p**	4-Ph	44.6 ± 1.9	48.4 ± 0.8	53.2 ± 2.5
**5q**	4-OCH_2_Ph	88.9 ± 2.1	93.6 ± 1.2	92.3 ± 2.6
**5r**	4-NCH_3_	12.3 ± 1.4	27.4 ± 3.5	34.2 ± 3.1
**5s**	3,5-CH_3_	17.3 ± 2.6	25.7 ± 3.0	36.4 ± 4.0
**5t**	3,5-OCH_3_	33.8 ± 1.8	35.5 ± 2.3	47.4 ± 2.9
SUT	-	83.8 ± 2.3	94.6 ± 1.9	94.6 ± 1.9

### 2.3. LPOR Inhibitory Experiments

To further demonstrate that the synthesized compounds functioned as LPOR inhibitors, an in vitro assessment of *At*LPOR inhibitory activity was conducted for three compounds, **5j**, **5k** and **5q**. The inhibitory effect on the *At*LPOR enzyme was evaluated by determining their IC_50_ values. As shown in Figure 3, the IC_50_ values of compounds **5j** (IC_50_ = 111.46 μM) and **5k** (IC_50_ = 83.63 μM) was much higher than that of compound **5q** (IC_50_ = 17.63 μM). And the IC_50_ value of compound **5j** was slightly higher than that of compound **5k**. Notably, compound **5q** exhibited the lowest IC_50_ value among all tested synthesized compounds, indicating the highest inhibition activity for *At*LPOR, consistent with the herbicidal activity data. These findings supported the view that compound **5q** functions as an LPOR inhibitor.

### 2.4. Crop Selectivity Test

To evaluate the postemergence crop safety of compound **5q**, trials were conducted on four representative crops: wheat, rice, corn and cotton. The results, as shown in Table 2 and Appendix A, indicate that at a dose of 150 g ai/ha, compound **5q** exhibited excellent safety for cotton, which is similar to SUT commercial herbicides. And the safety for rice is also similar to SUT, but both of them caused 40% injury. For wheat and corn, the damage rate of **5q** to crops is much greater than that of SUT. This suggests that compound **5q** may serve as a promising inhibitor for the management of weeds in cotton.

### 2.5. Molecular Docking

To gain insights into the inhibition mechanism of the highly active herbicidal compound at the molecular level, we performed molecular docking studies on the interaction between compounds **5j**, **5k** and **5q** and *At*LPOR. As depicted in Figure 4, all of the compounds could stably bind to the *At*LPOR protein. The binding mode of compounds **5q**, **5j** and **5k** with *At*LPOR involved hydrogen bonds and hydrophobic interactions, and compounds **5j** and **5k** have additional π–cation interactions with Arg121 (3.97 Å, 3.72 Å). However, Arg121 interacts with the five-membered heterocycle on compound **5j** and with the benzene ring on compound **5k**, which may be related to the position of the substituent NO_2_ on the benzene ring. In addition, compounds **5j** (Arg121, 2.91 Å; Arg317, 3.46 Å), **5k** (Asp146, 3.00 Å; Leu147, 2.31 Å) and **5q** (Thr313, 2.48 Å; Leu315, 3.48 Å) each have two hydrogen bond interactions. Furthermore, we can also see that compound **5q** has hydrophobic interactions with Ala174(3.43 Å), Val176(3.63 Å) and Thr197(3.56 Å), and these interactions also exist on compounds **5j** (Ala174, 3.34 Å; Val176, 3.97 Å) and **5k** (Val176, 3.66 Å; Thr197, 3.63 Å). As for why compound **5q** exhibits better herbicidal activity than compounds **5j** and **5k**, it is probably because compound **5q** has more hydrophobic interactions [25].

### 2.6. Molecular Dynamic Simulation

To further verify the reliability of screening and predict the stability of the inhibitor binding to the protein, a stable equilibrium system was obtained through a 100 ns simulation, and an RMSD curve (root mean square deviation), RMSF curve (root mean square fluctuation) and the contribution of amino acid residues to energy were analyzed. As shown in Figure 5A, both *At*LPOR–ligand complexes fluctuated within the range of 0.15 nm. Among them, the RMSD of *At*LPOR -**5q** was lower and fluctuated less, tending to be more stable. This showed that **5q** reaches equilibrium more quickly, and the overall conformation is more stable. The RMSF curves of the **5q**-*At*LPOR complex fluctuate within 1 nm in Figure 5B, showing that adding compound **5q** has little effect on the stability of the amino acid residues in the LPOR protein, suggesting that the complex formed is very stable. Furthermore, according to the contribution of amino acid residues to energy in Figure 5C, compound **5q** formed interactions with multiple amino acid residues in the *At*LPOR protein, among which Leu15 (−1.63494 Kcal/mol), Arg36 (−1.45541 Kcal/mol), Ala89 (−1.15884 Kcal/mol), Val91 (−0.9007 Kcal/mol) and Leu230 (−1.97915 Kcal/mol) played critical roles in binding. These key interactions enhanced its binding affinity with the active pocket. So, compound **5q** demonstrates stable binding to *At*LPOR, correlating with its enhanced herbicidal activity.

## 3. Materials and Methods

### 3.1. Equipment and Materials

All reagents were commercially available and used as purchased without further purification. The progress of the reactions was monitored by thin-layer chromatography (TLC). The melting points of the target compounds were measured on an RY-1G melting point meter (Tianguang Optical Instruments Co., Tianjin, China). NMR spectra were recorded on Bruker 500 MHz spectrometers (Brüker, Fällanden, Switzerland), in Chloroform-d (CDCl_3_). High-resolution mass spectrometry (HRMS) data were obtained using ultrahigh-performance liquid chromatography coupled with high-resolution mass spectrometry on a Q-Exactive plus system (Thermo Fisher, Waltham, MA, USA). Chemical shifts were reported in parts per million (ppm) relative to tetramethylsilane (TMS) as the internal standard. Enzyme activity tests were performed on the enzyme labeling instrument Multiskan Mk3 (Thermo, Shanghai, China).

### 3.2. Synthesis

#### 3.2.1. General Procedures for Synthesis of Compound **2**


Hydroxylammonium hydrochloride (1.68 g, 24.23 mmol) and Na_2_CO_3_ (1.64 g, 15.51 mmol) were dissolved by heating them in 100 mL water. Nitrile compounds (9.69 mmol) were mixed with ethanol (30 mL), and this solution was added to the former solution. The mixture was refluxed for 8 h. After cooling the reaction mixture to room temperature, the solvent was removed in vacuo. The residue was extracted with dichloromethane (3 × 10 mL), washed with water, dried over sodium sulfate, and concentrated in vacuo. Hexane was added to the residue, and the target compound 2 was isolated as a white solid and dried at room temperature [26].

#### 3.2.2. General Procedures for Synthesis of Compound **4**

Compound **2** was dissolved in acetone (15 mL). 2-benzyloxyacetyl chloride (1.36 g, 7.35 mmol) was dissolved in acetone (30 mL), and this solution was added slowly to the solution of compound **2**, and the mixture was stirred at room temperature for 24 h. Then, acetone was evaporated, and the residue was washed with sodium bicarbonate solution (5 mL) and water (10 mL) to obtain compound 4 [27,28].

#### 3.2.3. General Procedures for Synthesis of Compound **5**

Compound **4** was refluxed for 6 h in toluene (20 mL). Then toluene was removed, and ethyl acetate was added to the residue. The solvent was evaporated in vacuo, and the residue was purified via preparative chromatography [PE:EA= (10:1, *v*/*v*)] to obtain the target compound **5** [29,30].

5-((benzyloxy)methyl)-3-(4-ethoxyphenyl)-1,2,4-oxadiazole (**5a**). Yield: 78%. White solid. m.p. 44−45 °C. ^1^H NMR (500 MHz, CDCl_3_) δ 7.98–7.91 (m, 2H), 7.36–7.19 (m, 5H), 6.94–6.84 (m, 2H), 4.68 (s, 2H), 4.64 (s, 2H), 3.99 (q, *J* = 7.0 Hz, 2H), 1.35 (t, *J* = 7.0 Hz, 3H). ^13^C NMR (500 MHz, CDCl_3_) δ 175.71, 168.19, 161.48, 136.58, 129.17, 128.64, 128.31, 128.21, 118.76, 114.77, 73.57, 63.64, 62.44, 14.75. HRMS (Dual ESI): Calcd for C_18_H_19_N_2_O_3_ [M + H]^+^ 311.1317. Found 223.1398.

5-((benzyloxy)methyl)-3-(4-chlorophenyl)-1,2,4-oxadiazole (**5b**). Yield: 78%. Yellow oily. ^1^H NMR (500 MHz, CDCl_3_) δ 8.01–7.85 (m, 2H), 7.42–7.18 (m, 7H), 4.69 (s, 2H), 4.64 (s, 2H). ^13^C NMR (500 MHz, CDCl_3_) δ 176.28, 167.67, 137.53, 136.47, 129.26, 128.87, 128.68, 128.38, 128.21, 125.04, 73.67, 62.38. HRMS (Dual ESI): Calcd for C_16_H_14_ClN_2_O_2_ [M + H]^+^ 301.0666. Found 301.0747.

5-((benzyloxy)methyl)-3-(3-chlorophenyl)-1,2,4-oxadiazole (**5c**). Yield: 63%. Yellow oily. ^1^H NMR (500 MHz, CDCl_3_) δ 8.01 (t, *J* = 1.9 Hz, 1H), 7.89 (dt, *J* = 7.7, 1.4 Hz, 1H), 7.42–7.18 (m, 7H), 4.69 (s, 2H), 4.64 (s, 2H). ^13^C NMR (500 MHz, CDCl_3_) δ 175.31, 166.41, 135.37, 133.94, 130.34, 129.18, 127.61, 127.32, 127.16, 126.59, 124.54, 72.61, 61.30. HRMS (Dual ESI): Calcd for C_16_H_13_N_2_O_2_ClNa [M + Na]^+^ 323.0563. Found 323.0566.

5-((benzyloxy)methyl)-3-(4-fluorophenyl)-1,2,4-oxadiazole (**5d**). Yield: 72%. Yellow oily. ^1^H NMR (500 MHz, CDCl_3_) δ 8.00 (dd, *J* = 8.7, 5.7 Hz, 2H), 7.43–7.17 (m, 5H), 7.06 (t, *J* = 8.7 Hz, 2H), 4.68 (s, 2H), 4.63 (s, 2H). ^13^C NMR (500 MHz, CDCl_3_) δ 176.19, 167.65, 165.69, 163.68, 136.51, 129.77, 129.70, 128.66, 128.36, 128.21, 122.81, 122.78, 116.20, 116.02, 73.64, 62.39. ^19^F NMR (500 MHz, CDCl_3_) δ -111.77 (d, *J* = 9.2 Hz). HRMS (Dual ESI): Calcd for C_16_H_13_FN_2_O_2_ [M + H]^+^ 285.0961. Found 285.1033.

5-((benzyloxy)methyl)-3-(4-fluorophenyl)-1,2,4-oxadiazole (**5e**). Yield: 61%. White solid. m.p. 134.4−135.8 °C.^1^H NMR (500 MHz, CDCl_3_) δ 8.44 (t, *J* = 1.8 Hz, 1H), 8.04 (dt, *J* = 7.9, 1.4 Hz, 1H), 7.80 (dt, *J* = 8.0, 1.5 Hz, 1H), 7.44–7.25 (m, 5H), 7.17 (t, *J* = 7.9 Hz, 1H), 4.76 (s, 2H), 4.71 (s, 2H).^13^C NMR (500 MHz, CDCl_3_) δ 176.38, 167.14, 140.27, 136.49, 136.32, 130.58, 128.70, 128.46, 128.39, 128.23, 126.65, 94.50, 77.43, 73.70, 62.42. ^19^F NMR (500 MHz, CDCl_3_) δ −108.20. HRMS (Dual ESI): Calcd for C_16_H_14_FN_2_O_2_ [M + H]^+^ 285.0961. Found 285.1036.

5-((benzyloxy)methyl)-3-(4-iodophenyl)-1,2,4-oxadiazole (**5f**). Yield: 71%. White solid. m.p. 39−40 °C.1H NMR (500 MHz, CDCl_3_) δ 7.75 (s, 4H), 7.37–7.20 (m, 5H), 4.71 (s, 2H), 4.66 (s, 2H).13C NMR (500 MHz, CDCl_3_) δ 176.28, 167.91, 138.18, 136.43, 129.04, 128.68, 128.22, 126.02, 98.14, 73.69, 62.38. HRMS (Dual ESI): Calcd for C_16_H_14_IN_2_O_2_ [M + H]^+^ 393.0022. Found 393.0100.

5-((benzyloxy)methyl)-3-(4-iodophenyl)-1,2,4-oxadiazole (**5g**). Yield: 82%. White solid. m.p. 134.4−135.8 °C.^1^H NMR (500 MHz, CDCl_3_) δ 7.80 (d, *J* = 7.8 Hz, 1H), 7.71 (d, *J* = 9.1 Hz, 1H), 7.41–7.18 (m, 6H), 7.11 (t, *J* = 8.4 Hz, 1H), 4.70 (s, 2H), 4.64 (s, 2H).^13^C NMR (500 MHz, CDCl_3_) δ 176.36, 167.62, 163.90, 161.94, 136.47, 130.68, 130.61, 128.67, 128.62, 128.55, 128.37, 128.22, 123.29, 123.26, 118.43, 118.27, 114.71, 114.52, 77.29, 73.67, 62.37. HRMS (Dual ESI): Calcd for C_16_H_14_IN_2_O_2_ [M + H]^+^ 393.0022. Found 393.0099.

5-((benzyloxy)methyl)-3-(4-bromophenyl)-1,2,4-oxadiazole (**5h**). Yield: 65%. Yellow oily. ^1^H NMR (500 MHz, CDCl_3_) δ 7.92–7.82 (m, 2H), 7.57–7.48 (m, 2H), 7.36–7.19 (m, 5H), 4.69 (s, 2H), 4.64 (s, 2H).^13^C NMR (500 MHz, CDCl_3_) δ 176.30, 167.76, 136.45, 132.22, 129.05, 128.69, 128.39, 128.23, 125.97, 125.48, 73.68, 62.38. HRMS (Dual ESI): Calcd for C_16_H_14_BrN_2_O_2_ [M + H]^+^345.0160. Found 345.0232.

5-((benzyloxy)methyl)-3-(3-bromophenyl)-1,2,4-oxadiazole (**5i**). Yield: 61%. White solid. m.p. 31−32 °C.^1^H NMR (500 MHz, CDCl_3_) δ 8.17 (s, 1H), 7.94 (d, *J* = 7.8 Hz, 1H), 7.53 (d, *J* = 7.0 Hz, 1H), 7.38–7.17 (m, 6H), 4.69 (s, 2H), 4.64 (s, 2H).^13^C NMR (500 MHz, CDCl_3_) δ 176.39, 167.35, 136.44, 134.34, 130.54, 130.49, 128.69, 128.46, 128.40, 128.23, 126.05, 123.00, 73.69, 62.38. HRMS (Dual ESI): Calcd for C_16_H_14_BrN_2_O_2_ [M + H]^+^ 345.0160. Found 345.0234.

5-((benzyloxy)methyl)-3-(4-nitrophenyl)-1,2,4-oxadiazole (**5j**). Yield: 77%. White solid. m.p. 84−85 °C. ^1^H NMR (500 MHz, CDCl_3_) δ 8.29–8.14 (m, 4H), 7.41–7.19 (m, 5H), 4.74 (s, 2H), 4.67 (s, 2H).^13^C NMR (500 MHz, CDCl_3_) δ 176.28, 167.67, 137.53, 136.47, 129.26, 128.87, 128.68, 128.38, 128.21, 125.04, 73.67, 62.38. HRMS (Dual ESI): Calcd for C_16_H_14_N_3_O_4_ [M + H]^+^ 312.0906. Found 312.1007.

5-((benzyloxy)methyl)-3-(3-nitrophenyl)-1,2,4-oxadiazole (**5k**). Yield: 66%. White solid. m.p. 42−43 °C.^1^H NMR (500 MHz, CDCl_3_) δ 8.85 (t, *J* = 2.0 Hz, 1H), 8.38–8.20 (m, 2H), 7.59 (t, *J* = 8.0 Hz, 1H), 7.37–7.19 (m, 5H), 4.74 (s, 2H), 4.67 (s, 2H).^13^C NMR (500 MHz, CDCl_3_) δ 176.93, 166.88, 148.65, 136.36, 133.11, 130.13, 128.70, 128.43, 128.33, 128.23, 125.87, 122.64, 77.31, 73.79, 62.36. HRMS (Dual ESI): Calcd for C_16_H_14_N_3_O_4_ [M + H]^+^312.0906. Found 312.0978.

5-((benzyloxy)methyl)-3-phenyl-1,2,4-oxadiazole (**5l**). Yield: 85%. Yellow oily. ^1^H NMR (500 MHz, CDCl_3_) δ 8.09–7.95 (m, 2H), 7.48–7.16 (m, 8H), 4.68 (s, 2H), 4.63 (s, 2H).^13^C NMR (500 MHz, CDCl_3_) δ 174.97, 167.38, 135.48, 130.27, 127.83, 127.57, 127.24, 127.13, 126.48, 125.48, 72.52, 61.34,. HRMS (Dual ESI): Calcd for C_16_H_15_N_2_O_2_ [M + H]^+^ 267.1055. Found 267.1127.

5-((benzyloxy)methyl)-3-(4-(methylthio)phenyl)-1,2,4-oxadiazole (**5m**). Yield: 61%. White solid. m.p. 44−45 °C.^1^H NMR (500 MHz, CDCl_3_) δ 7.97–7.86 (m, 2H), 7.37–7.19 (m, 7H), 4.70 (s, 2H), 4.65 (s, 2H), 2.44 (s, 3H).^13^C NMR (500 MHz, CDCl_3_) δ 175.94, 168.13, 143.18, 136.52, 128.66, 128.35, 128.22, 127.80, 125.87, 122.79, 73.62, 62.42, 15.08. HRMS (Dual ESI): Calcd for C_17_H_17_N_2_O_2_S [M + H]^+^ 313.0932. Found 313.1006.

5-((benzyloxy)methyl)-3-(p-tolyl)-1,2,4-oxadiazole (**5n**). Yield: 83%. Yellow oily. ^1^H NMR (500 MHz, CDCl_3_) δ 7.94–7.82 (m, 2H), 7.34–7.11 (m, 7H), 4.66 (s, 2H), 4.61 (s, 2H), 2.29 (s, 3H).^13^C NMR (500 MHz, CDCl_3_) δ 174.77, 167.33, 140.61, 135.45, 128.53, 127.55, 127.22, 127.13, 126.38, 122.59, 72.47, 61.33, 20.51. HRMS (Dual ESI): Calcd for C_17_H_17_N_2_O_2_ [M + H]^+^ 281.1212. Found 281.1273.

5-((benzyloxy)methyl)-3-(4-methoxyphenyl)-1,2,4-oxadiazole (**5o**). Yield: 70%. Yellow oily. ^1^H NMR (500 MHz, CDCl_3_) δ 8.02–7.89 (m, 2H), 7.37–7.19 (m, 5H), 6.96–6.86 (m, 2H), 4.69 (s, 2H), 4.65 (s, 2H), 3.78 (s, 3H).^13^C NMR (500 MHz, CDCl_3_) δ 174.68, 167.10, 161.02, 135.51, 128.13, 127.59, 127.26, 127.17, 117.93, 113.25, 72.52, 61.38, 54.35. HRMS (Dual ESI): Calcd for C_17_H_17_N_2_O_3_ [M + H]^+^ 297.1161. Found 297.1237.

3-([1,1′-biphenyl]-4-yl)-5-((benzyloxy)methyl)-1,2,4-oxadiazole (**5p**). Yield: 82%. White solid. m.p. 81−82 °C. ^1^H NMR (500 MHz, CDCl_3_) δ 8.08 (d, *J* = 6.9 Hz, 2H), 7.62 (d, *J* = 8.0 Hz, 2H), 7.54 (d, *J* = 7.6 Hz, 2H), 7.46–7.17 (m, 8H), 4.70 (s, 2H), 4.65 (s, 2H). ^13^C NMR (500 MHz, CDCl_3_) δ 176.08, 168.29, 144.12, 140.16, 136.56, 128.97, 128.69, 128.37, 128.25, 128.03, 127.59, 127.21, 125.39, 73.65, 62.47. HRMS (Dual ESI): Calcd for C_22_H_19_N_2_O_2_ [M + H]^+^ 343.1368. Found 343.1464.

5-((benzyloxy)methyl)-3-(4-(benzyloxy)phenyl)-1,2,4-oxadiazole (**5q**). Yield: 64%. White solid. m.p. 134.4−135.8 °C.^1^H NMR (500 MHz, CDCl_3_) δ 8.07–7.99 (m, 2H), 7.49–7.25 (m, 10H), 7.09–7.01 (m, 2H), 5.10 (s, 2H), 4.76 (s, 2H), 4.71 (s, 2H).^13^C NMR (500 MHz, CDCl_3_) δ 175.78, 168.14, 161.26, 136.59, 136.46, 129.23, 128.72, 128.67, 128.34, 128.21, 127.55, 119.25, 115.21, 73.59, 70.11, 62.45. HRMS (Dual ESI): Calcd for C_23_H_21_N_2_O_3_ [M + H]+ 373.1474. Found 373.1548.

4-(5-((benzyloxy)methyl)-1,2,4-oxadiazol-3-yl)-*N,N*-dimethylaniline (**5r**). Yield: 64%. White solid. m.p. 67−68 °C.^1^H NMR (500 MHz, CDCl_3_) δ 7.94–7.81 (m, 2H), 7.44–7.19 (m, 5H), 6.71–6.58 (m, 2H), 4.67 (s, 2H), 4.63 (s, 2H), 2.94 (s, 6H).^13^C NMR (500 MHz, CDCl_3_) δ 175.24, 168.56, 152.28, 136.67, 133.42, 128.76, 128.63, 128.27, 128.23, 113.64, 111.69, 111.43, 73.52, 62.51, 40.16. HRMS (Dual ESI): Calcd for C_18_H_20_N_3_O_2_ [M + H]^+^ 310.1477. Found 310.1561.

5-((benzyloxy)methyl)-3-(3,5-dimethylphenyl)-1,2,4-oxadiazole (**5s**). Yield: 72%. Yellow oily. ^1^H NMR (500 MHz, CDCl_3_) δ 7.68–7.60 (m, 2H), 7.36–7.19 (m, 5H), 7.05 (s, 1H), 4.70 (s, 2H), 4.64 (s, 2H), 2.29 (s, 6H).^13^C NMR (500 MHz, CDCl_3_) δ 175.86, 168.64, 138.62, 136.55, 133.07, 128.66, 128.33, 126.23, 125.27, 73.62, 62.45, 21.22. HRMS (Dual ESI): Calcd for C_18_H_18_N_2_O_2_ [M + H]^+^ 295.1368. Found 295.1441.

5-((benzyloxy)methyl)-3-(3,5-dimethoxyphenyl)-1,2,4-oxadiazole (**5t**). Yield: 88%. White solid. m.p. 134.4−135.8 °C.1H NMR (500 MHz, CDCl_3_) δ 7.48–7.14 (m, 7H), 6.58 (q, *J* = 2.2 Hz, 1H), 4.76 (s, 2H), 4.70 (s, 2H), 3.81 (d, *J* = 2.2 Hz, 6H).13C NMR (500 MHz, CDCl_3_) δ 175.78, 168.14, 161.26, 136.59, 136.46, 129.23, 128.72, 128.67, 128.34, 128.21, 127.55, 119.25, 115.21, 73.59, 70.11, 62.45. HRMS (Dual ESI): Calcd for C_18_H_19_N_2_O_4_ [M + H]^+^ 327.1267. Found 327.1344.

### 3.3. Herbicidal Activity Assay

We tested the postemergence herbicidal activities [31,32,33] of two weed species: *Arabidopsis thaliana* (AT) and *Digitaria sanguinalis* (DS). Using dimethyl sulfoxide (DMSO) as the solvent and Tween-80 as the emulsifier, all tested compounds were formulated into a 20 mg/mL emulsion concentrate for later use, which was diluted with water to the desired concentration during the experiment. Uniformly grown weeds were selected as the test subjects, and the herbicidal activity of AT was tested at the doses of 37.5, 75 and 150 g ai/ha, and sulfentrazone (SUT) was selected as the positive control. Subsequently, three compounds with the best herbicidal activity were selected and tested for the herbicidal activity of DS at a dose of 150 g ai/ha. The experimental procedures were conducted strictly following previously reported methods. After 14 days, the herbicidal efficacy of the compounds against the target weed species was visually assessed.

### 3.4. Determination of LPOR Enzyme Inhibitory Activity in Vitro

The Plant LPOR ELISA Kit (JLC11293) from Shanghai Jingkang Bioengineering Co., Ltd. was utilized to evaluate the in vitro inhibitory activity of compounds **5j**, **5k** and **5q** on the LPOR enzyme. The determination of LPOR enzyme inhibitory activity in vitro was performed according to a reported method [13]. The procedure was as follows [34,35]: First, 0.3 g of *Arabidopsis thaliana* leaves was taken, and they were immersed in 3 mL of cold phosphate-buffered saline (PBS, pH 7.4) for washing. Excess liquid was gently removed from the surface using an absorbent paper. PBS was added at a weight-to-volume ratio of 1:9 (g/mL), and the plant leaves were thoroughly crushed in a mortar on ice before being centrifuged at 1000× *g* for 20 min using a low-temperature freezer centrifuge to collect the upper layer liquid. Subsequent experiments were performed according to the instructions provided with the kit. A standard curve was plotted based on the results of the standard samples, and the enzyme concentrations at different concentrations of various compounds were determined. The half maximal inhibitory concentration (IC_50_) for each compound was calculated, with assays performed in triplicate.

### 3.5. Crop Selectivity

To assess whether the candidate compounds cause harm to crops, wheat, rice, corn and cotton were selected as test crops to evaluate crop selectivity under greenhouse conditions. The application rates for the candidate compounds were set at 150 g ai/ha, with SUT serving as the control. Crop safety was assessed 7 days after application, with each experiment replicated three times.

### 3.6. Molecular Docking

The structure of compounds **5j**, **5k** and **5q** was built and optimized in SYBYL 2.1. Charges for compounds were calculated using the Gasteiger–Huckel method, which was optimized for 5000 steps using the Powell method, with the optimization terminating when the energy change was less than 0.005 kcal/mol. The crystal structure of *At*LPOR (PDB ID: 7JK9) was retrieved from the RCSB Protein Data Bank and prepared for molecular docking. The docking of the compounds to *At*LPOR was performed using the Docking Suite module in SYBYL 2.1. And the interactions between the compounds and receptor were visualized using PyMOL software [36,37,38].

### 3.7. Molecular Dynamic Simulation

Molecular dynamic simulation experiments [39,40] were conducted using Amber 2022 software to investigate the interaction between compounds and target proteins. The system used an ff14SB force field for 3D protein construction and gaff2 for 3D ligand construction. The molecular dynamic simulation system was filled using TIP3P-type water molecules, and Na^+^ and Cl^−^ ions were added according to the system charge for equilibration. Energy minimization, limiting ligands, system equilibrium, finished product simulation and result analysis were performed subsequently. The simulated system was operated at a pressure of 1 bar and a temperature of 300 K for 100 ns. The molecular dynamic simulation trajectory data were analyzed by using the cpptraj module of Amber 2022. In addition, we applied the “MM/PBSA” script to calculate the molecular mechanics/Poisson−Boltzmann surface area (MM/PBSA) and determine the binding free energy of protein–inhibitor complexes.

## 4. Conclusions

In summary, we developed a series of 1,2,4-oxadiazole compounds (compounds **5a** to **5t**) as novel LPOR inhibitors. A systematic herbicidal activity evaluation of 20 synthesized compounds revealed that at a dosage of 150 g ai/ha, compounds **5j**, **5k** and **5q** exhibited more than 90% inhibition against AT and DS, comparable to the commercial formulation SUT. In vitro inhibition experiments indicated that compared with compounds **5j** and **5k**, compound **5q** exhibited higher efficacy. Crop safety tests indicated that compound **5q** can be applied at 150 g ai/ha in cotton. Molecular docking analysis highlighted significant hydrogen bonds and hydrophobic interactions between compound **5q** and amino acid residues. Furthermore, molecular dynamic simulations further illustrated that compound **5q** and protein LPOR are stably bound. These experimental and theoretical results suggest that compound **5q** has the potential to become a lead compound for LPOR inhibitor herbicides.

## Data Availability

Data are contained within the article and Appendix A.

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
