# Peer review of "Design, Synthesis and Herbicidal Activity of 1,2,4-Oxadiazole Compounds as Novel Light-Dependent Protochlorophyllide Oxidoreductase Inhibitors"

_molecules, 2025, doi:10.3390/molecules30193970_

Round 1
Reviewer 1 Report
Comments and Suggestions for Authors
This article describes the synthesis of new 1,2,4-oxadiazole compounds (20 new ones). Each compound was characterized by NMR, and spectra were provided. Herbicidal properties were determined for some of the organic molecules obtained. The article is eligible for publication in the journal Molecules after some revisions.
The main question for the entire article is: Why is 5q so superior in a seemingly similar series of compounds? Did the authors test the solubility of the compounds they obtained in the medium in which they studied their activity? And also the stability of the solution, at least using electronic absorption spectra recorded over time?
“low-toxicity pesticides9”
“less than 20 kinds of targets10”
Line 85-87: “It is worth mentioning that hydroxylammonium hydrochloride and 85 Na2CO3 were dissolved in water at first.” - How does hydroxylammonium hydrochloride affect the solubility of sodium carbonate? Is it possible that the hydrochloric acid in hydroxylammonium hydrochloride reacts with sodium carbonate?
Line 89-90: “More importantly, we need to make sure that when synthesizing compound 2, com-pound 1 is fully reacted, otherwise it will be difficult to separate compound 1 in the final purification step.” - This is indeed true, it is not entirely clear why this proposal is made, because it is obvious
With all due respect to the authors, the reaction scheme needs to include not only the reaction conditions, but also the reagents (or describe what R is, and make a visual table in this scheme).
Line 97: “All figures and tables should be cited in the main text as Figure 1, Table 1, etc.”
Lines 118-119: “ However, it is worth noting that when the substituent is OCH2Ph (5q), the herbicidal activity is very good.” - In general, the initial choice of substituents is interesting; there could have been hundreds of them. Why did the authors settle on these ones?
Lines after 154: “As for why compound 5q exhibits better herbicidal activity than compounds 5j and 5k, probably because compound 5q has more hydrophobic interactions.” - This is an interesting hypothesis. Could the authors provide examples from the literature where hydrophobic interactions have influenced the increase in herbicidal activity?
Line 181: “As illustrated in Scheme 1, compound 2 was dissolved in acetone (15 mL).” - With all due respect to the authors, this is not indicated in the Scheme.
Line 187: “As illustrated in Scheme 1, compound 4 was refluxed for 6 h in toluene (20 mL). Then toluene was removed and ethyl acetate was added to the residue.” - With all due respect to the authors, this is not indicated in the Scheme
5n if I’m not mistaken can be find here:
Reaxys ID: 53150180
US2006/14945, 2006, A1
Current Patent Assignee: HOFFMANN LA ROCHE
https://ppubs.uspto.gov/pubwebapp/external.html?q=(20060014945).pn.&db=USPAT,US-PGPUB
Reviewer 2 Report
Comments and Suggestions for Authors
This manuscript reports the design and synthesis of a new series of 1,2,4-oxadiazole compounds targeting light-dependent protochlorophyllide oxidoreductase as herbicides. The authors synthesized 20 compounds, evaluated post-emergence herbicidal activity in greenhouse assays, measured in vitro LPOR inhibition, assessed crop selectivity, and performed molecular docking and dynamics simulations. The data suggest that 5q could be a lead for developing novel LPOR-inhibiting herbicides. In summary, this manuscript presents a novel and well-executed study on LPOR-inhibiting herbicidal compounds with promising results. The experimental design is strong and the conclusions are supported by the data. Once the authors address the points raised the manuscript should be suitable for publication in Molecules.
Specific comments
- One suggestion is that the manuscript would benefit from reporting actual percentage inhibition or biomass reduction values (with error bars) rather than only categorical ratings (“+” signs). Quantitative data (means ± SD) and statistical analysis of the differences would strengthen the evidence and allow readers to assess variability.
- Compound 5q caused considerable injury to wheat and corn, indicating its selectivity may be limited to certain crops. The authors might expand the discussion of these results by commenting on why cotton might tolerate 5q while other crop species do not.
- The description of the in vitro LPOR inhibition experiment needs more clarity. The authors used a “Plant LPOR ELISA Kit” on Arabidopsis thaliana leaf extracts to determine IC50 values. It is not immediately clear how an ELISA kit was used to measure enzyme activity in the presence of inhibitors. The authors should clarify the assay principle in the Methods. If available, providing a brief description of the calculation of IC50 or citing a reference for the assay method would be helpful.
- The IC50 values for compounds are presented without error bars or standard deviations. Since these values are central to the claim that 5q is the most potent inhibitor, the authors should report variability (e.g., mean ± SD from at least three replicates) and ideally include error bars in Figure 3.
- The manuscript should clarify the weed species used in the herbicidal assays and ensure consistent nomenclature. The text refers to “broadleaf weeds (AR)” and uses the abbreviation AT in the Methods. Similarly, “grass weed (DS)” is mentioned without defining the species. The authors must explicitly state the scientific or common names of the weed species.
- The authors mention using the MM/PBSA method to calculate binding free energies, but no binding energy results are given in the text. If the authors decided not to include these results, they might consider removing mention of the MM/PBSA from methods to avoid raising expectations.
- Many sentences require minor editing for grammar or clarity. Ensure all acronyms are defined at first use.
